# A Meta-Analysis of the Effects of Different Training Modalities on the Inflammatory Response in Adolescents with Obesity

**DOI:** 10.3390/ijerph192013224

**Published:** 2022-10-14

**Authors:** Haotian Zhao, Ruihong Cheng, Jin Teng, Ge Song, Chenjian Huang, Shuo Yuan, Yuxuan Lu, Siqin Shen, Jingqi Liu, Chang Liu

**Affiliations:** 1Department of Physical Education, Jiangnan University, Wuxi 214122, China; 2School of Sports Engineering, Beijing Sport University, Beijing 100084, China; 3School of Sport Science, Beijing Sport University, Beijing 100084, China; 4Faculty of Sports Science, Ningbo University, Ningbo 315211, China; 5Faculty of Engineering, University of Pannonia, 8200 Veszprém, Hungary

**Keywords:** training modalities, inflammation, meta-analysis, obese, adolescents

## Abstract

The aim of this study was to investigate the effect of different training modalities on improving the inflammatory response in adolescents with obesity. For the study methodology, the databases such as China National Knowledge Infrastructure (CNKI), Wanfang Data, Pubmed, Web of Science, and EBSCO were selected for searching. The methodological quality of the included studies was assessed using the Cochrane Risk of Bias (ROB) tool, and statistical analysis was performed by applying RevMan 5.4.1 analysis software. A total of 14 studies with 682 subjects were included. The results of this meta-analysis showed that aerobic training (AT) and aerobic plus resistance training (AT + RT) reduced the levels of IL-6 and CRP in adolescents with obesity. Among them, AT + RT was more effective than other training modalities in reducing IL-6 and CRP in adolescents with obesity. Different training modalities have no effect on the TNF-α level in adolescents with obesity. However, regarding the increase in IL-6, CRP, and TNF-α in adolescents with obesity, resistance training (RT) did not lead to significant differences. In conclusion, long-term regular AT, AT + RT, and HIIT are all helpful in improving the inflammatory state of adolescents with obesity, with AT + RT being the best training modality to combat inflammation in adolescents with obesity.

## 1. Introduction

Obesity is a complex problem that affects children of all age groups. According to the World Health Organization (WHO) and the World Obesity Federation (WOF), more than 150 million children and adolescents between the ages of 5 and 19 were suffering from childhood obesity worldwide by 2020; by 2030 this number could exceed 250 million [1]. Numerous studies have confirmed that obesity is closely related to risk factors for cardiovascular disease. More than 67% of adolescents with obesity have at least one cardiovascular risk factor, while more than 25% of children and adolescents with obesity have two or more cardiovascular risk factors [2]. Obesity increases the risk of early puberty in children, menstrual irregularities in adolescent, sleep disorders such as obstructive sleep apnea, cardiovascular risk factors, diabetes mellitus (including prediabetes type 2), high cholesterol levels, hypertension, nonalcoholic fatty liver disease, and metabolic syndrome. In addition, children and adolescents who are obese may suffer from psychological problems such as depression, anxiety, poor self-esteem, body image, and peer relationships, and eating disorders [3].

To date, interventions to prevent overweight/obesity in children have concentrated on individual behavior changes, such as increasing daily physical activity or enhancing diet quality by limiting excessive calorie intake [4]. Infrequently trained adolescents are prone to obesity, leading to a chronic inflammatory response throughout the whole body [5,6]. Currently, most children and adolescents in the world do not meet the recommendation of moderate to high-intensity physical activity for at least 60 min per day [7]. Training is, however, effective in alleviating obesity and suppressing the inflammatory response in skeletal muscle [8].

Although training has been shown to be effective in reducing the acute phase protein C-reactive protein (CRP) and cytokines such as tumor necrosis factor alpha (TNF-α) and interleukin 6 (IL-6), there is no consensus on the improvement of CRP, IL-6, and TNF-α in adolescents with obesity through different training modalities [9]. These results may be due to the enhanced antioxidant capacity of skeletal muscle in all different age groups of trainers, leading to enhanced anti-inflammatory capacity through the upregulation of superoxide dismutase (SOD), catalase (CAT), and glutathione peroxidase (GPX) [10].

Specifically, CRP is an acute phase protein that is synthesized by hepatocytes in response to pro-inflammatory cytokines during inflammation/infection. It is known as a biomarker of acute inflammation, but many large prospective studies have shown that CRP is also associated with chronic inflammation, such as cardiovascular disease, type 2 diabetes, age-related macular degeneration, hemorrhagic stroke, Alzheimer’s disease, and Parkinson’s disease [11]. TNF-α was initially thought to contribute to tumor necrosis, but it has recently been found to have additional important functions as a pathological component of autoimmune diseases. These pathways lead to a variety of cellular responses, including cell survival, differentiation, and proliferation. However, inappropriate or excessive activation of the TNF-α signaling pathway is associated with chronic inflammation, which may ultimately lead to the development of pathological complications, such as autoimmune diseases [12]. IL-6, a member of the pro-inflammatory cytokine family, induces the expression of various proteins responsible for acute inflammation and plays an important role in the proliferation and differentiation of human cells. IL-6 is produced rapidly and transiently in response to infection and tissue injury and promotes host defense by stimulating acute phase responses and hematopoietic and immune responses. Although its expression is tightly controlled by transcriptional and post-transcriptional mechanisms, the persistent dysregulation of IL-6 synthesis has a pathological role in chronic inflammation and autoimmunity [13].

The different training modalities mainly include aerobic training, resistance training, aerobic plus resistance training, and HIIT. “Aerobic” is defined as “involving, relating to, or requiring oxygen” and refers to the adequate use of oxygen to meet energy requirements through aerobic metabolism during training. Aerobic training refers to a series of trainings that rely primarily on aerobic energy-production processes, which are accomplished by repeating a series of light-to-moderate-intensity activities over a long period of time. Resistance training involves physical training activities designed to improve strength and endurance. It is usually associated with weight lifting and includes various training techniques such as callisthenic, isometric, and plyometric. High-intensity interval training (HIIT) is a training regimen that alternates short periods of high-intensity or explosive anaerobic training with short recovery periods until depletion, thus relying almost maximally on the anaerobic energy release system [14].

This meta-analysis aimed to compare the effects of different training modalities on the inflammatory response level in adolescents with obesity, to provide a reliable theoretical reference for the rational training modalities selection, and to improve the related diseases caused by the systemic chronic inflammatory response in adolescents with obesity.

## 2. Materials and Methods

### 2.1. Search Strategy

This study was conducted strictly in accordance with the preferred reporting items for systematic reviews and meta-analysis (PRISMA) guidelines. For the literature search, the keywords such as “training intervention”, “aerobic training”, “resistance training”, “aerobic combined with resistance training”, “high-intensity interval training”, “adolescents with obesity”, and “inflammation”, “inflammatory response”, and “inflammatory factors” were searched in China National Knowledge Infrastructure, Wanfang Data, PubMed, Web of Science, and EBSCO. The search strategy combined subject terms and free words and tracked the relevant literature references. The search date was from the beginning of the database creation to January 2022.

### 2.2. Selection and Exclusion Criteria of the Literature

Inclusion criteria: (1) subjects: the subjects were overweight and adolescents with obesity (13–18 years old), and the criteria for overweight and obesity were based on accepted standards of the World Health Organization (WHO) or the country where the subjects were located; (2) study type: randomized controlled trial or self-controlled trial; (3) the subjects had obesity but without any metabolic disorder or cardiovascular disease; (4) outcome measures included IL-6, TNF-α and CRP.

Exclusion criteria: (1) adolescents with average weight or adolescents of different ages; (2) the literature with no full text or unclear experimental data; (3) conference abstracts, dissertations, case studies, reviews, and other gray literature; and (4) duplicate publications.

### 2.3. Study Selection and Data Extraction

Endnote citation management software was used to assist in screening the literature. The extracted content mainly included the first author of the literature, the year of publication, the sample size, the number of men and women, the age, the form of training, the intervention method (training modalities, periodicity, duration, frequency), and the outcome evaluation metrics. The literature screening and data extraction were carried out independently by two researchers (HTZ, JT) and cross-checked with each other after completion. If there was any disagreement, it was submitted to a third researcher (SQS) for further discussion and decision.

### 2.4. Quality Assessment

The Cochrane Risk of Bias (ROB) tool in RevMan5.4.1 analysis software was used to evaluate the literature quality in six indicators: selectivity bias, measurement bias, implementation bias, loss of follow-up bias, and reporting bias. The evaluation results included low risk, uncertainty, and high risk. If the six indicators were all low-risk, the studies were defined as low-risk and high-quality studies; if one or more indicators are uncertain, studies are defined as medium-quality studies; if one or more indicators were high-risk, studies were defined as high-risk and low-quality studies. The quality assessment was carried out independently by two researchers (HTZ, JT). In the case of disagreement, it was referred to a third researcher (SQS) for discussion and decision.

### 2.5. Statistical Analysis

The data were analyzed using RevMan5.4.1(Cochrane, London, UK). Effect sizes were expressed as weighted mean differences (WMD), and 95% CI was calculated. The heterogeneity was analyzed using the consistency coefficients I^2^ and *p*. If I^2^ > 50%, *p* < 0.10, it proved that each study had heterogeneity, and the random effect model was used for analysis; on the contrary, the fixed effect model was used. Subgroup analyses were performed according to the training modality. Sensitivity analyses were used to detect the effect of individual studies on the total effect. When statistical heterogeneity existed but there was no clinical heterogeneity between study groups, a random effect model was used for analysis. If the heterogeneity was too pronounced to identify its source, only descriptive analyses were used. When the number of studies was ≥10, funnel plots were used to explore publication bias and α = 0.05 was chosen as the significance test level of the articles.

## 3. Results

### 3.1. General Results of the Selected Research Literature

A search and screening of 682 publications was first conducted, resulting in the inclusion of 14 studies. Figure 1 shows the specific literature screening process.

### 3.2. General Features of the Selected Research Literature

The basic characteristics of the included studies are shown in Table 1. A total of 14 randomized controlled trials (RCTs) included 781 subjects. Seven studies included the AT group, two studies included the RT group, nine studies included the AT + RT group, and only one study included the HIIT group. By country, South Korea and Brazil each had four studies, while Singapore, Canada, the United States, Germany, Serbia, and China each had one study. For the experimental design component, different numbers of adolescents with obesity were recruited by means including but not limited to flyers or school-based directories and then randomly assigned to either the intervention or the control group for training. The intervention group participated in different training programs according to the experimental design, while the control group maintained the same lifestyle but without training intervention and subsequently measured various inflammatory response indicators, including CRP, TNF-α, or IL-6. The gender and age of the subjects remained consistent, i.e., the subjects and controls had a balanced gender ratio and were of similar age, as shown in Table 1.

### 3.3. Quality Evaluation of the Selected Literature

According to the Cochrane Risk of Bias Tool 2.0, the quality and bias risk of 14 included studies were evaluated, including two low-risk studies, seven medium-risk studies, and five high-risk studies, as shown in Figure 2.

### 3.4. Meta-Analysis of the Effect of Different Training Modalities on the Level of IL-6 in Adolescents with Obesity

Regarding the effect of different training modalities on IL-6 levels in adolescents with obesity, there was homogeneity among studies (I^2^ = 0%, *p* = 0.59). Using the fixed-effect model, the combined effect amount SMD = −0.28, 95% CI = −0.62~0.06 (*p* = 0.10), indicating that training can reduce the level of IL-6 in adolescents with obesity, but the result was not significant. Subgroup analysis showed that in AT (SMD = −0.27, 95% CI = −0.72~0.19, *p* = 0.25) and AT + RT (SMD = −0.67, 95% CI = −1.38~0.03, *p* = 0.06) could reduce the level of IL-6 (Figure 3).

### 3.5. Meta-Analysis of The Effects of Different Training Modalities on the Level of Tnf-A in Adolescents with Obesity

Regarding the effect of different training modalities on TNF-α levels in adolescents with obesity, there was homogeneity between these studies (I^2^ = 45%, *p* = 0.11). Using a fixed-effect model, the combined effect size SMD = 0.34, 95% CI = −0.03~0.65 (*p* = 0.03), indicating that training could not reduce the TNF-α in adolescents with obesity. Subgroup analysis showed that AT (SMD = 0.36, 95% CI = −0.11~0.82, *p* = 0.13), RT (SMD = −0.83, 95% CI = 0.08~1.58, *p* = 0.03), and AT + RT (SMD = 0.34, 95% CI = −0.39~0.60, *p* = 0.69) could not reduce the TNF-α level in adolescents with obesity (Figure 4).

### 3.6. Meta-Analysis of the Effect of Different Training Modalities on the Level of CRP in Adolescents with Obesity

Regarding the effect of different training modalities on CRP levels in adolescents with obesity, there was heterogeneity between the studies (I^2^ = 76%, *p* < 0.00001). A random-effect model was adopted. The combined effect size SMD = −0.40, 95% CI = −0.68~−0.12 (*p* = 0.005), indicating that training significantly reduced the level of CRP in adolescents with obesity. Subgroup analysis showed that AT (SMD = −0.35, 95% CI = −0.67~−0.03, *p* = 0.03) and AT + RT (SMD = −0.60, 95% CI = −1.10~−0.10, *p* < 0.00001) significantly reduced CRP levels (Figure 5).

### 3.7. Publication Bias and Sensitivity Analysis

The funnel plot was drawn for the indicators of IL-6, TNF-α, and CRP. The results showed that the distribution of the funnel plots of IL-6 and TNF-α was largely symmetrical, while the distribution of the funnel plots of CRP was largely asymmetrical (Figure 6, Figure 7 and Figure 8). The sensitivity analysis was carried out by varying the combined effect size. For the above indicators, individual studies were excluded before the meta-analysis was performed again. The results showed that the changes were not significant compared with the previous results, indicating that the meta-analysis results were stable and that the conclusions of this study were reliable.

## 4. Discussion

A total of 14 studies with 682 subjects were included in this study. A meta-analysis of the subjects’ inflammatory response data showed that training improved IL-6 levels and significantly reduced CRP levels in adolescents with obesity, but training did not improve TNF-α levels in adolescents with obesity. Subgroup analysis of the different training modalities showed that AT + RT was superior to AT, RT, and HIIT, which is consistent with the study by Welsh C et al., [29]. Since AT + RT is superior to AT, RT, and HIIT alone in reducing body fat and regulating lipid metabolism, AT achieves a reduction in body fat percentage by reducing fat mass, while RT achieves a reduction in body fat percentage by increasing muscle weight [30]. In addition, the effect of aerobic and resistance training on body composition was superior to that of single training, suggesting that combined training has a better effect on body composition than single training. Combined training may better improve body composition and thus improve the chronic inflammatory state caused by obesity in adolescents with obesity [31].

The high heterogeneity of IL-6 and CRP in the current study, as well as the variety of movements, forms, and training environments involved in the training interventions, make it possible for the results to be biased by other factors, and the gender ratio in the included relevant studies was asymmetrical. Meanwhile, the study by IRENA et al., also showed significant differences in the levels of IL-6 and CRP between genders and that changes in the level of inflammatory response to training after the intervention were also uncertain [32]. In addition, most of the included studies did not explicitly describe the diet of the subjects during the training intervention. These may have contributed to the high heterogeneity of the studies.

AT and AT + RT were superior to RT in reducing IL-6 levels in adolescents with obesity, which is similar to previous studies [33,34] and may be due to resistance training-induced muscle micro damage as well as exercise-induced fatigue [35]. It has been shown that training-induced muscle damage can persist for 24–72 h after resistance training [35], and in the RT studies included in this meta-analysis, most observations were made at the time point of 48 h after the last exercise session, when serum IL-6 had not returned to quiet values.

In terms of TNF-α, we found that all these four training modalities failed to reduce TNF-α levels in adolescents with obesity. In addition to the time point of observation after the last training session mentioned above, the intervention period of the included studies may have contributed to the lack of training effect. The intervention period of the studies included in this meta-analysis ranged from 6 to 12 weeks, and some studies showed that it took at least 1 year for training to cause a significant decrease in TNF-α levels [36,37].

Finally, as far as CRP is concerned, we found that AT + RT is superior to other forms of training. Unlike IL-6 and TNF-α, CRP is an acute phase plasma protein that is synthesized not only by the liver but also by adipocytes [38]. AT + RT not only improves CRP levels by regulating lipid metabolism through low body fat, but also stimulates skeletal muscle metabolism by activating the adenosine monophosphate-activated protein kinase (AMPK) signaling pathway [39], helping to regulate hepatic glycogen synthesis and lipolysis, slowing muscle loss, and increasing body metabolism throughout the day, thereby improving inflammation levels.

## 5. Limitations of Current Research

We strictly followed the prescribed procedures of PRISMA, but due to the small number of included HIIT studies, there was some heterogeneity in the findings.

Adolescents with obesity tend to be in a state of chronic inflammation due to high body fat content, which may interfere with the assessment of changes in inflammatory factors. Training interventions involve multiple movements and forms and the training environment is variable, which may bias the results due to other factors.

## 6. Conclusions

The AT, RT, and AT + RT all improved BMI in adolescents with obesity while reducing levels of the inflammatory response factor CRP; AT + RT was more effective than other training modalities in reducing the inflammatory response factor CRP in adolescents with obesity. Our results showed that training was effective in reducing the IL-6 levels, which confirms the effectiveness and scientific validity of training in alleviating chronic low-grade inflammation in the adolescent population with obesity. Subgroup analysis showed that AT + RT was the best training modality to improve the inflammatory response in adolescents with obesity, and future researchers can use AT + RT as an entry point to explore the optimal intervention cycle, intervention intensity, and intervention frequency based on gender.

## Figures and Tables

**Figure 1 ijerph-19-13224-f001:**
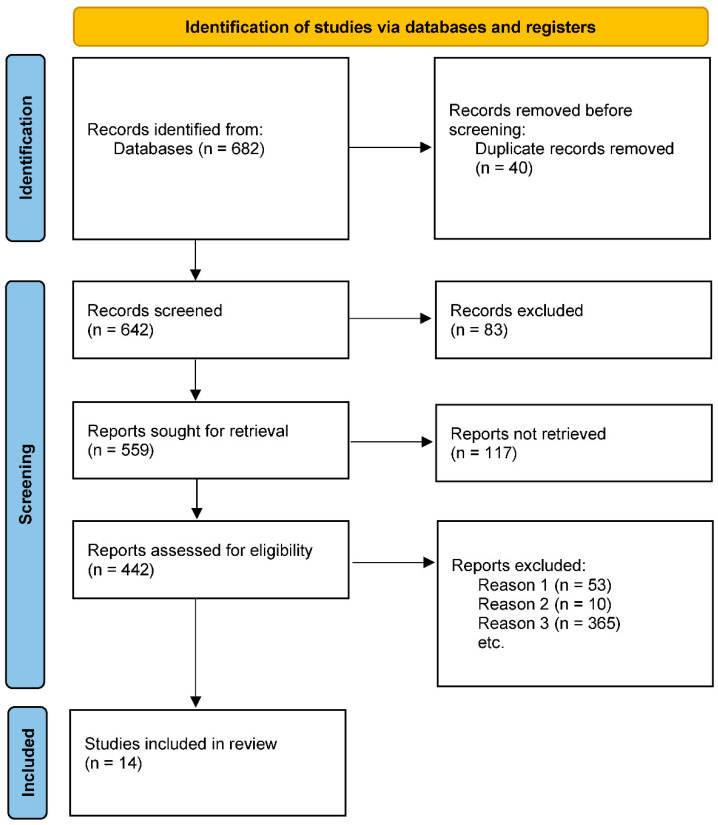
Flow chart of literature screening.

**Figure 2 ijerph-19-13224-f002:**
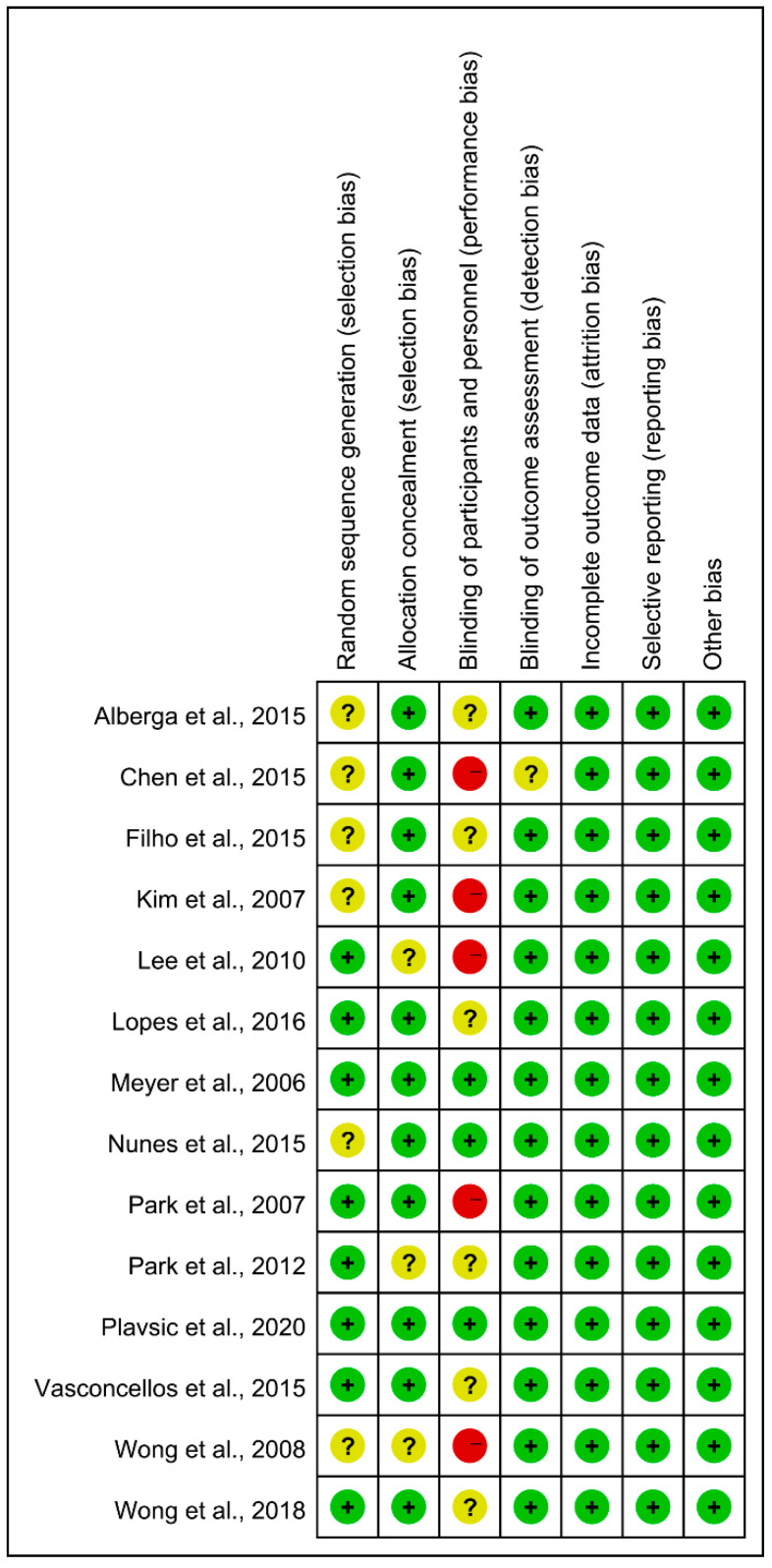
Risk of bias summary. Note: +—Low risk of bias; −—High risk of bias; ?—Unclear risk of bias.

**Figure 3 ijerph-19-13224-f003:**
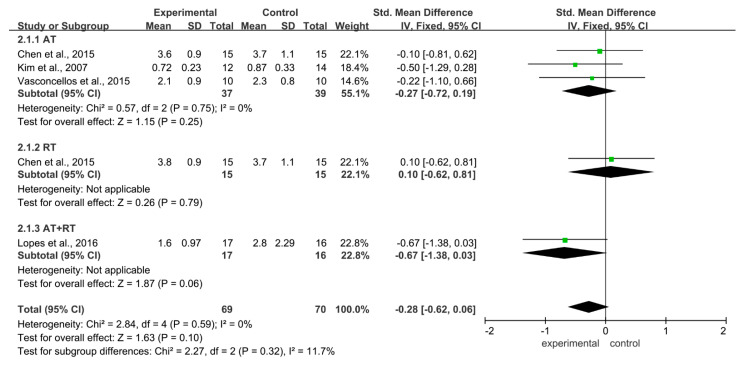
Meta-analysis of the effect of different training modalities on the level of IL-6 in adolescents with obesity.

**Figure 4 ijerph-19-13224-f004:**
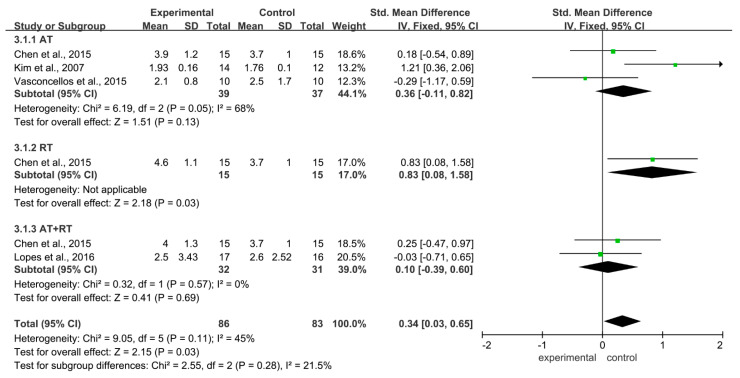
Meta-analysis of the effects of different training modalities on the level of TNF-α in adolescents with obesity.

**Figure 5 ijerph-19-13224-f005:**
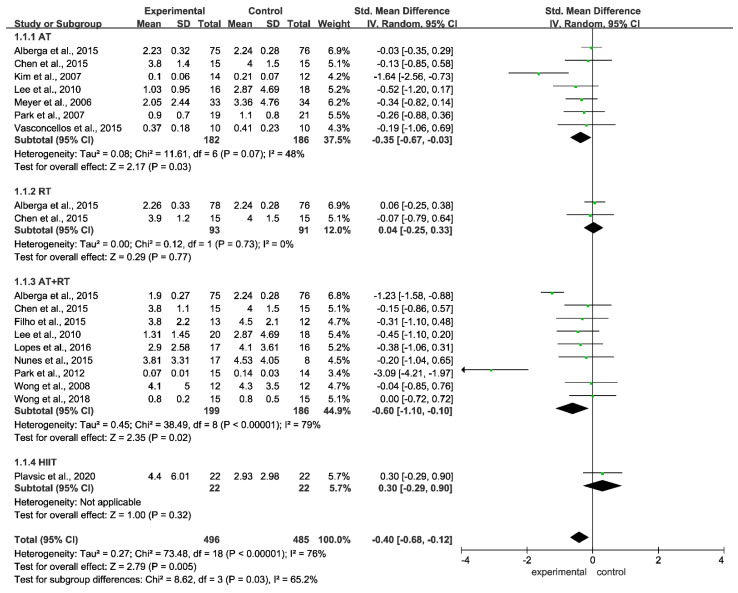
Meta-analysis of the effect of different training modalities on the level of CRP level in adolescents with obesity.

**Figure 6 ijerph-19-13224-f006:**
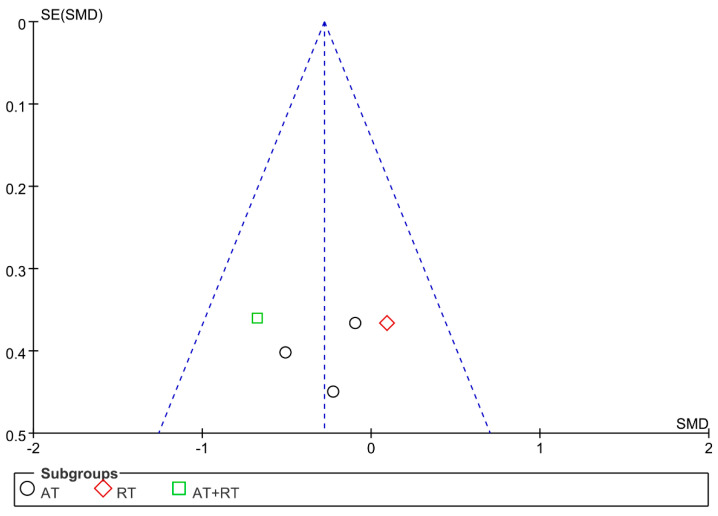
Publication bias of the influence of different exercise modalities on the level of IL-6 in adolescents with obesity.

**Figure 7 ijerph-19-13224-f007:**
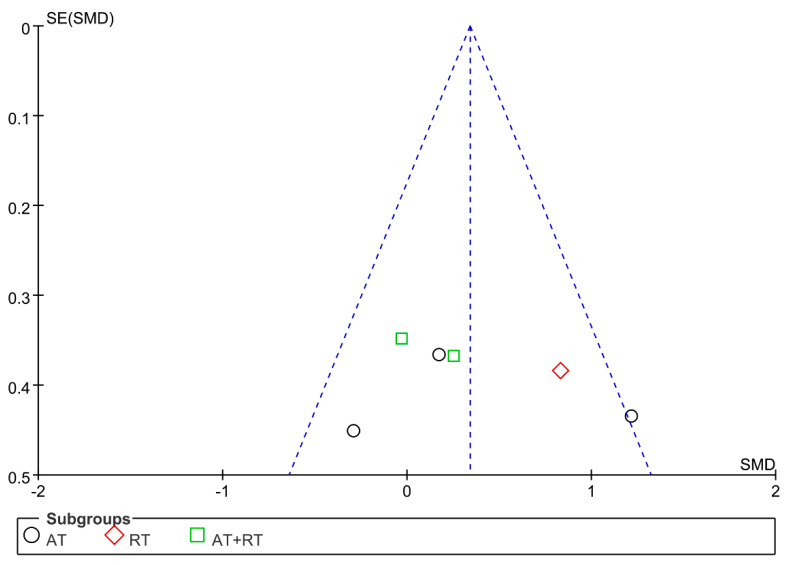
Publication bias of the influence of different exercise modalities on the level of TNF-α in adolescents with obesity.

**Figure 8 ijerph-19-13224-f008:**
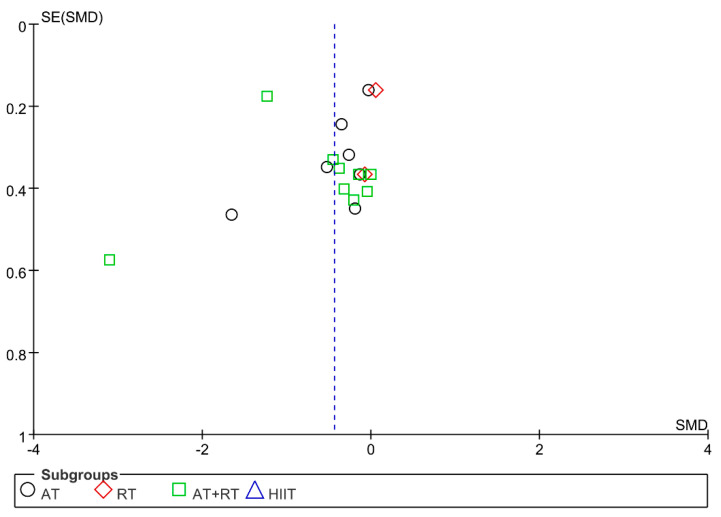
Publication bias of the influence of different training modalities on the level of CRP in adolescents with obesity.

**Table 1 ijerph-19-13224-t001:** General features of the selected research literature.

Author	Year	Research Subject		Training Mode Set			Outcomes
		Age	Sample Size (Male/Female)	Training Mode	Training Frequency (n/Week)	Training Duration (Week)	
Park et al., [15]	2007	AT:14.2 ± 0.5	19 (0/19)	AT	6	12	CRP
		C: 14.1 ± 0.5	21 (0/21)				
Kim et al., [16]	2007	AT: 17 ± 0.11	14 (14/0)	AT	5	6	CRP, TNF-α, IL-6,
		C: 17 ± 0.11	12 (12/0)				
Wong et al., [17]	2008	AT + RT: 13.75 ± 1.06	12 (12/0)	AT + RT	2	12	CRP
		C: 14.25 ± 1.54	12 (12/0)				
Alberga et al., [18]	2015	AT: 15.5 ± 1.4	75 (22/53)	AT	4	22	CRP
		RT: 15.9 ± 1.5	78 (23/55)	RT			
		AT + RT: 15.5 ± 1.3	75 (22/53)	AT + RT			
		C: 15.6 ± 1.3	76 (24/52)				
Park et al., [19]	2012	AT + RT: 12.1 ± 0.1	15 (7/8)	AT + RT	3	12	CRP
		C: 12.2 ± 0.1	14 (7/7)				
Wong et al., [20]	2018	AT + RT: 15.2 ± 1.2	15 (0/15)	AT + RT	3	12	CRP
		C: 15.3 ± 1.1	15 (0/15)				
Nunes et al., [21]	2015	AT + RT: 16.18 ± 1.51	17 (8/9)	AT + RT	:2	24	CRP
		C: 15.4 ± 1.2	8 (4/4)				
Vasconcellos et al., [22]	2015	AT: 14.1 ± 1.3	10 (8/2)	AT	3	12	CRP, TNF-α, IL-6
		C: 14.8 ± 1.4	10 (6/4)				
Lopes et al., [23]	2016	AT + RT: 14.6 ± 1.15	17 (0/17)	AT + RT	3	12	CRP, TNF-α, IL-6,
		C: 14.4 ± 1.16	16 (0/16)				
Meyer et al., [24]	2006	AT: 13.7 ± 2.1	33 (17/16)	AT	3	24	CRP
		C: 14.7 ± 2.2	34 (17/17)				
Lee et al., [25]	2010	AT: 13 ± 1.0	16 (45/9)	AT	3	10	CRP
		AT + RT: 13 ± 1	20	AT + RT			
		C: 13 ± 1	18				
Filho et al., [26]	2015	AT + RT: 13.4 ± 1.3	13 (6/7)	AT + RT	3	8	CRP
		C: 13.7 ± 0.9	12 (6/6)				
Plavsic et al., [27]	2020	HIIT: 15.5 ± 1.5	22 (0/22)	HIIT	2	12	CRP
		C: 16.2 ± 1.3	22 (0/22)				
Chen et al., [28]	2015	AT: 14.1 ± 3.1	15 (15/0)	AT	3	8	CRP, TNF-α, IL-6,
		RT: 13.9 ± 2.2	15 (15/0)	RT			
		AT + RT: 14.2 ± 3.8	15 (15/0)	AT + RT			
		C: 14.4 ± 3.2	15 (15/0)				

C—control; CRP—C-reactive protein; TNF-α—tumor necrosis factor alpha; IL-6—interleukin 6.

## Data Availability

Not applicable.

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
