# Peer review of "A Meta-Analysis of the Effects of Different Training Modalities on the Inflammatory Response in Adolescents with Obesity"

_ijerph, 2022, doi:10.3390/ijerph192013224_

Round 1
Reviewer 1 Report
The author conducted a meta-analysis of the effects of different exercise modes on the Inflammatory response of obese adolescents.
The authors should explain in more detail why they targeted obese adolescents.
The analysis of CRP had serious heterogenity (I'square was very high). The authors should explain it in Discussion section.
Why did the authors show funnel plot in CRP? They should show the publication bias of IL-6 and TNF-a.
Author Response
Thanks for the reviewer’s professional comments and suggestions.
Please see the attachment.
Thank you

Reviewer 2 Report
Thank you for the opportunity to review the paper. I think this paper focuses on an important topic; however, study quality is undermined by the unclear writing, lack of details (especially in the method section), and inconsistency between results and conclusion.
General comments
1. Please use “people-first language”, avoiding saying “obese children…”
2. Writing needs to be improved—there are grammatical errors throughout the paper, and sentences did not flow well
Abstract
1. Please spell out the abbreviations
2. More details are needed, e.g., study type? How many exercise modes were examined? What inflammatory biomarkers were included?
3. The description of “Long-term regular AT, AT + RT, and HIIT…” is vague. What defines long-term?
Introduction
1. The prevalence of pediatric obesity was cited from 2020 reports, so past tense should be used
2. Any previous studies that compared the effect of different exercise modes on inflammatory responses among adults? If so, those studies should be discussed.
3. What are the possible mechanisms underlying exercise and reduced inflammation?
Methods
1. Please specify what “other databases” were searched
2. Was grey literature searched?
3. One of the inclusion criteria was “subjects were healthy”—this is a very vague description. Does it mean that subjects were “metabolically healthy” (subjects had obesity but without any metabolic disorder or cardiovascular disease)?
4. Any criteria on sample size?
5. Why were only three inflammatory biomarkers selected? What about IL-1α, IL-1β, and IL-12?
6. Were review papers excluded?
7. Not sure why future tense was used (e.g., shall be…)
8. More details are needed for the paper selection process. Who did the screening? How many abstracts and full papers were screened?
Results
1. Please give a brief description of included studies (e.g., country distribution, design, control group) and subjects (e.g., sex, mean age)
Discussion
1. In the results section, the authors reported that the effect of exercise on IL-6 and TNF- α was not significant, which contradicts their conclusion “This study found that exercise can improve the IL-6 and TNF- α of obese adolescents”
2. This study is focused on children and adolescents, but the authors mentioned “elderly” in the discussion?
3. A missing piece in the discussion is that different inflammatory biomarkers may respond differently to exercise and weight loss interventions
4. What are the limitations of the included studies, as well as this review?
5. What are the implications for future studies?
Conclusion
The conclusion is not supported by the study findings.
Author Response
Thanks for the reviewer’s professional comments and suggestions.
Please see the attachment.
Thank you.

Reviewer 3 Report
The aim of the authors was to investigate the effects of different exercise modes on improving in-flammatory response in the obese adolescents. A total of 14 studies (20 randomized controlled trials and 11 self-controlled trials) with 682 subjects were included. The results of this meta-analysis showed that AT and AT + RT could reduce the levels of IL-6 and CRP in obese adolescents. Among them, the effect of AT + RT on reducing IL-6 (SMD = -0.67, 95% CI = -1.38-0.03, P = 0.06) and CRP (SMD = -0.30, 95% CI = -0.90-0.29, P = 0.32) of obese adolescents was better than that of other ex-ercise methods; Effects of different exercise modes on TNF- α level in obese adolescents has no effect.
The authors concluded that Long-term regular AT, AT + RT, and HIIT are all helpful to improve the inflam-matory state of obese adolescents, among which AT + RT is the best exercise mode for obese ado-lescents to resist inflammation.
The manuscript is interesting and well written.
These are my comments with a pure academic spirit:
1. The abstract must be revised. It must better summarize the sections and written with more care see for example “methods; Effects”
2. Insert the acronyms in a tabe
3. Improve the resolution of the figures.
4. Insert the limitations encountered in the studies in the discussion
5. Insert the limitations of your study in the discussion
Author Response

(The authors gave the same response as above.)

Round 2
Reviewer 1 Report
The authors responded appropriately.
Author Response
We are very grateful to the reviewers for their positive comments and suggestions, and we will continue to work hard to live up to the reviewers' support in the future.
Reviewer 2 Report
Thank you for the authors' careful responses. The quality of the manuscript has improved; however, more work is still needed to enhance the writing and readability (e.g., obese adolescents were used throughout the paper)
Author Response
We would like to express our highest appreciation and respect to the reviewer for his/her valuable comments and suggestions once again.
Under his/her guidance, we once again invited a professional English editing agency to re-touch our manuscript to improve our English.We also double-checked our manuscript according to "people-first language". Specifically, we changed "obese children" to "children with obesity" and "obese adolescents" to "adolescents with obesity" according to the language guidelines of "World Obesity Healthy Voices"(https://www.worldobesity.org/downloads/healthy_voices_downloads/HV_Language_guidelines.pdf). (lines 3, 15, 21, 22, 23, 24, 26, 27, 28, 32, 35, 36, 41, 47, 49, 56, 92, 94, 102, 108, 114, 161, 178, 183, 188, 191, 195, 198, 200, 202, 207, 213, 225, 228, 231, 235, 236, 245, 255, 263, 279, 284, 286, 290).
We are grateful for the reviewers' comments, which helped us to make the manuscript scientifically rigorous, unbiased, and even humanistic in its language.
